# Cyber Resilience Progression Model

**Juan F. Carías** [1],*[ID]**, Saioa Arrizabalaga** [1,2]**, Leire Labaka** [1][ID] **and Josune Hernantes** [1]

1   School of Engineering, TECNUN, University of Navarra, 20018 San Sebastian, Spain;
    sarrizabalaga@ceit.es (S.A.); llabaka@tecnun.es (L.L.); jhernantes@tecnun.es (J.H.)
2   CEIT, 20018 San Sebastian, Spain
*   Correspondence: jfcarias@tecnun.es

**Abstract:** Due to the hazardous current cyber environment, cyber resilience is more necessary than ever. Companies are exposed to an often-ignored risk of suffering a cyber incident. This places cyber incidents as one of the main risks for companies in the past few years. On the other hand, the literature meant to aid on the operationalization of cyber resilience is mostly focused on listing the policies required to operationalize it, but is often lacking on how to prioritize these actions and how to strategize their implementation. Therefore, the usage of the current literature in this state is not optimal for companies. Thus, this study proposes a progression model to help companies strategize and prioritize cyber resilience policies by proposing the natural evolution of the policies over time. To develop the model, this study used semi-structured interviews and an analysis of the data obtained from the interviews. Through this methodology, this study found the starting points for each cyber resilience policy and their natural progression over time. These results can help companies in their cyber resilience building process by giving them insights on how to strategize the implementation of the cyber resilience policies.

**Keywords:** cyber resilience; maturity model; progression model

## 1. Introduction

The consistently growing number of cyberattacks and variety of cyber threats through the previous years' reports [1–3] demonstrates the risk that the current cyber scenario represents for companies. Moreover, these cyber threats are being targeted towards small and medium-sized enterprises (SMEs) because these represent a significant payload as a group and often have limited knowledge and resources to face these threats [4]. This cyber threat scenario makes cyber incidents one of the most impactful and one of the most probable risks that companies face [5,6].

On the other hand, technology is advancing fast, making the cyber threat scenario more volatile and facing it even harder. For instance, Industry 4.0 introduces the challenge of connecting OT technologies into IT-like networks, which has been studied because of the complexity that this adds when trying to protect networks and systems [7,8]. The aggressive cyber threat scenario combined with the rapid changes in technology has made the task of creating protected and fail-safe systems (the objective of traditional cybersecurity [9]) an unrealistic way of protecting a company from cyber incidents. In this sense, several experts and entities [10–12] have agreed that cybersecurity requires a broader approach that they have called cyber resilience.

Cyber resilience is defined as the "ability of a process, business, organization, or nation to anticipate, withstand, recover, and evolve in order to improve their capabilities in face of adverse conditions, stress, or attacks to the cyber resources it needs to function" [13]. Some authors also include the ability to detect in this definition [12,14]. This concept, contrary to traditional cybersecurity, intends to make companies and their systems safe-to-fail [9]. This apparently slight change in the point

of view converts cybersecurity in a much more robust strategy that is prepared to respond, recover and adapt from incidents, which traditionally has not been in the priorities of cybersecurity [15,16]. Thus, cyber resilience can also make companies more robust against the changing cyber threat scenario, with more robust and adaptable ways to face cyber threats [11].

Although implementing a cyber resilience point of view can be a solution for companies to face the current evolution of technology and the aggressive cyber threat scenario, cyber resilience can prove to be hard to operationalize. This difficulty stems from its comprisal of several domains (governance, business continuity, information security, etc.) with hundreds of policies within them [14,17,18] and intricate relationships between these domains and policies [19]. Thus, the problem considered in this article is that cyber resilience operationalization is difficult but needed by companies in the current cyber scenario.

The current literature tries to ease the process of operationalizing cyber resilience through frameworks, self-assessment questionnaires, standards and maturity models [13,14,18,20–25]. A current literature review identified over 200 cyber resilience assessment frameworks [26]. However, these frameworks often include very detailed lists of policies and actions without means for prioritization of these actions or strategies on how to implement each action in the company. Most frameworks can serve as examples of enumeration of policies and domains that companies should implement in order to operationalize cyber resilience [12,14,17,27]. For instance, one of the most popular cybersecurity frameworks is the National Institute of Standards and Technology's (NIST's), which lists over 100 policies (called subcategories) in over 30 domains (called categories) to achieve cyber resilience. The document in which these policies are defined explicitly says that they should be selected by the company according to a previous profiling [14], but the framework has no means, instructions or resources on how to do this profiling and how to select and prioritize these policies once the profiling is done.

Another example of enumeration in the current literature is metrics which usually list ways to measure an underlying set of policies and domains [28–30]. For instance, the MITRE corporation's set of cyber resilience metrics contains over 200 metrics and recommends companies to use as few as possible since metrics need to be interpreted and the less they are the easier it is to understand their values [28]. However, this document leaves the selection and prioritization of these metrics to the companies' judgment.

Similar to metrics, self-assessment questionnaires often give insight on how the company is now based on an underlying set of policies that companies should follow [13,18,25,31]. Sometimes, these tools can also give suggestions on actions that the company could do to improve its current cyber resilience, but these suggestions are also based on the underlying policies and, therefore, also need prioritization. For instance, the assessment tool proposed by Benz and Chatterjee is based on NIST's framework and its suggestions are to improve the shortcomings of the company by complying with an associated NIST subcategory [25]. This results in a list of subcategories from NIST's framework that are more specific to the company's situation. However, in companies starting their cyber resilience operationalization process this list might still be extensive and require prioritization and customization of those recommendations.

Standards can also be used to aid in cyber resilience operationalization [20,32]. The most known example of a standard in this field is the ISO 27000, which can be summarized as a guide on how to make an Information Security Management System (ISMS) and use it to manage information security in a company. Like the ISO 27000, which focuses on information security (a part of cyber resilience [24]), most standards focus on a single aspect of cyber resilience rather than give companies a holistic approach. For companies starting their cyber resilience operationalization, this require looking for different standards for different parts of cyber resilience. Standards also include several actions and processes that should be implemented in the companies that wish to be certified on that standard. This means that all the policies in a standard should be implemented, but like in the previous approaches, a standard does not give companies starting their cyber resilience operationalization a

way to adapt those processes and actions to their own situation. These processes and actions can also require prioritization since standards often have several of these and, depending on the company's circumstances, some may be more important than others.

Finally, as tools to aid companies in cyber resilience operationalization, the literature has maturity models [18,21,33,34], which are in essence sets of characteristics that define a development in a certain entity or field put sequentially in a limited number of stages or levels [35,36]. Capability maturity models are designed to measure and describe how mature companies' processes are and how embedded these processes are in the company's culture [35]. Thus, it is not a detailed guideline on how to start to operationalize but rather a way of improving or implementing processes that help companies internalize cyber resilience. In practice, this also means that companies require the knowledge to implement these processes and thus the policies and domains supporting them.

Although it is reasonable to require a profiling of the companies' circumstances or customization of the provided tools, it is also true that many companies will not be able to prioritize correctly or that will require more knowledge, experience and investment in order to do so. Thus, these types of documents can overwhelm companies starting their cyber resilience operationalization process and, therefore, there is a need for guidelines and other kind of material to help companies operationalize cyber resilience based on the information already available on actions and policies. Therefore, the closest to guidelines on how to implement cyber resilience in the current literature are maturity models. Nonetheless, there are three types of maturity models: progression models, capability maturity models, and hybrid models [35] and the current literature offers only capability maturity models [13,18,21,31] which, as mentioned before, require implemented processes and policies to be used effectively by companies.

Progression models, on the other hand, are descriptions of natural progressions over time of characteristics, attributes or policies, which makes their main purpose to provide roadmaps or guidelines expressed as better versions of these policies as the scale progresses [35]. This kind of model can be a better starting point for companies to operationalize cyber resilience, since it describes an implementation from its most basic state, which may be more attainable than achieving a capability or process maturity state when there is no current implementation of the characteristics, attributes or policies in question.

Thus, the goal of this article is to create a progression model based on the most essential cyber resilience policies and domains in order to serve as a guideline for companies starting their cyber resilience operationalization process. In order to do this, 11 experts participated in semi-structured interviews to define the progression of 33 cyber resilience policies within 10 domains. This progression model can be useful for SMEs or companies starting their operationalization process to have a guideline on how to implement cyber resilience policies by prioritizing them according to their maturity level.

The rest of this article describes in detail the process by which the progression model are defined in Section 2. Subsequently, the progression model is given in Section 3. Then, a discussion on the usage, limitations and future lines research is given in Section 4. Finally, the main conclusions of this article are given in Section 5.

## 2. Methodology

This study used semi-structured interviews as a source of qualitative information in order to obtain a cyber resilience progression model. Semi-structured interviews are a means of data collection for qualitative research useful to gather information about a particular topic or area from the experience of individuals [37]. In this particular case, this methodology was used to collect information on the progression of cyber resilience policies from experts of three different profiles: cybersecurity providers (3), cybersecurity researchers (3), and professionals in companies in charge of cybersecurity implementation (5). These experts were selected due to their wide experience in the field and their profiles were chosen to add the three perspectives on the topic. These three perspectives were considered to add value to the study because academia is rigorous but often disconnected from the empirical experience of practitioners, organization practitioners often have the empirical

experience but can sometimes lack the most recent advances in the field, and cybersecurity providers are dedicated to implementing cyber resilience policies in companies in their daily lives so they might have a combination of both of the others. This diversity of backgrounds also reaffirmed the usage of semi-structured interviews as a way to ensure a standardized understanding of the questions and vocabulary in the interview [38].

On the other hand, previous research has defined 10 domains and 33 policies as the most essential cyber resilience domains and policies [24,39]. This work takes these as the base to build the cyber resilience progression model. Using these findings, the semi-structured interviews were designed and later quantified and analyzed in order to define how these policies progressed over time. The design of the interviews, their development, and their analysis are described in the next subsections.

### 2.1. Interviews' Design and Execution

In order to obtain a progression model from the experts' point of view, the interviews were designed to be a systematic construction of a progression model. To achieve this, all the experts were given a simplified version of the domains and policies (e.g., "make an inventory of assets" instead of "Make an inventory that lists and classifies the company's assets and identifies the critical assets"). These simplifications were made to avoid biasing the perspective of the expert by adding advanced characteristics of the policy within the way of writing it. The table with the domains and simplified policies was given to the experts in a document that served as the interviews' script. This document also contained the definitions for "cyber resilience" and "progression model" as well as the objectives of the interviews, the expected results and the following two-step methodology:

Step1: Establish a starting point for each cyber resilience policy. In this step, they were also asked to keep in mind dependencies among these policies and their own experience in order to place these policies on a starting point from a scale of 1–5. Where one is the least advanced, least mature of companies and five the most advanced maturity level. The scale was selected based on other maturity models in the literature, which vary from three to six maturity levels [18,21,31].

Step 2: Describe how these policies progress over the next steps of the scale. For instance, if a policy starts at level three, how the policy manifests in a company at level three, then level four, and finally level five.

In these steps the scale was defined without names for each maturity level to avoid biasing through the usage of names (e.g., if level 5 is called "Optimizing" as in the Capability Maturity Model Integration (CMMI) [40], the actions described in level five would be limited to optimization actions). Other maturity models in the literature also avoid the usage of names for their maturity levels [18,21].

With this information, the experts were interviewed one by one and the 11 interviews were recorded to ensure the correct transcription of the interviews. The transcriptions were also sent to each expert in order to double check that their ideas were captured accurately and avoid incorrect recording of data and/or a biased interpretation of what the experts responded. A graphical overview of how the resulting progression model from each expert is shown in Figure 1. This is also similar to the final result of this article after the analysis of the interviews described in the following subsection.

| Domain | Policy | 1 | 2 | 3 | 4 | 5 |
|---|---|---|---|---|---|---|
| Domain 1 | Policy 1 | | Policy starting description | Evolution description at level 3 | Evolution description at level 4 | Evolution description at level 5 |
| | Policy 2 | | | Policy starting description | Evolution description at level 4 | Evolution description at level 5 |
| | Policy 3 | | Policy starting description | Evolution description at level 3 | Evolution description at level 4 | Evolution description at level 5 |
| Domain 2 | Policy 4 | | | | Policy starting description | Evolution description at level 5 |
| | Policy 5 | | Policy starting description | Evolution description at level 3 | Evolution description at level 4 | Evolution description at level 5 |
| | Policy 6 | Policy starting description | Evolution description at level 2 | Evolution description at level 3 | Evolution description at level 4 | Evolution description at level 5 |

**Figure 1.** Graphical overview of resulting progression models.

During the interviews, the experts commonly asked the following questions:

- If they could start a policy at level five and, therefore, have no evolution. This was allowed as it was considered an interesting statement on the complexity of a policy.
- If policies could stay the same through various levels (or skip them, which was equivalent) and evolve when the level of maturity was higher (for example, stay the same from 1–3 and change in 4 and 5). This kind of evolution was also allowed since it would allow a realistic view of the progression of a policy.
- If they should try to depict reality or define the best possible scenario. In this case, they were asked to do their best to be realistic but in case they believed a policy is not applied in their context to try to place the policy in an ideal starting point considering the companies' capacities at that maturity level.

To avoid fatigue during the interviews and thus bias due to this fatigue, the interviews were limited as much as possible to one hour and 30 min. The average duration of the interviews was 1 h and 20 min. This was possible because the experts were given the scripts previously and the transcription of their ideas was made after the interview, which permitted the experts to speak freely and without delays. Moreover, the experts were given freedom to choose the order of the policies but 73% of them decided to use the order in the underlying framework from [24,39] shown to them in the script. Nevertheless, there were no noticeable signs of fatigue during the interviews.

*2.2. Analysis of Interview Transcripts*

As mentioned in the previous subsection, at the end of the interviews, the transcripts resulted in progression models from the point of view of each expert. To analyze these transcripts, the five-step methodology for analyzing semi-structured interviews suggested by Schmidt was used [41]. These five steps are:

1. Material-oriented formation of analytical categories, which consists of carefully reading and understanding the individual transcripts. In this step, annotations were made on the common concepts found among the transcripts as they were read individually.
2. Assembly of the analytical categories into a guide for coding, which in the case of this study consists of creating categories that summarize the different types of progression identified by the experts. These types of progression were defined by grouping common patterns on the experts' progressions (found on the interviews) and naming these patterns as descriptively as possible. Table 1 defines the progression types found during this step and later used for the coding step.
3. Coding of the material, which consisted of assigning a progression type to each of the policies and progressions from the transcript of each expert. In this step, multiple codes could be assigned to each policy's progression from a single expert.
4. Quantifying surveys of material, which consisted of two parts: (1) determining whether there was consensus on the starting maturity of each policy and (2) determining whether there was consensus on the progression type for each policy.

**Table 1.** Progression types for coding and their definitions.

| Category | Code | Definition |
|---|---|---|
| Investment Increase | II | This code was assigned when the expert's progression description was related to an increase in the resources (mainly economic resources) dedicated to implementing/operationalizing the policy. |
| Continuity | C | This code was assigned when the expert's progression description was based on the increase of frequency in which the policy's actions are performed in the company (i.e., it was done more and more frequently as the level increased). |
| Specificity | S | This code was assigned when the expert's progression describes an increase in level of detail in which the policy is done as the maturity of the company increases. (i.e., it started in a general way and became more detailed and specific as the level increased). |
| Expansion | E | This code was assigned when the expert's progression description included the expansion of the policy's action in the company (e.g., the action was performed in some sections of the company and it started being done in more sections as the level increased). |
| Formalization | F | This code was assigned when the expert's progression description referred to the documentation or systematization of the actions (i.e., when the policy's actions started being intuitive or informal and where standardized and documented as the level increased). |
| Independence | I | This code was assigned when the expert's progression description mentioned the decrease of dependency of the company from the help of cybersecurity providers or external entities to perform the tasks related to the policy. |
| Optimization | O | This code was assigned when the expert's progression description was based on the measurement and improvement of Key Performance Indicators (KPIs) to optimize the performance of the policy's actions. |
| Proactivity | P | This code was assigned when the expert's progression description represented a change of attitude from the company towards the policy's actions (e.g., from complying to pursuing it for their own perceived benefit). The mention of continuous improvement in actions that could not be quantified was coded under this category as well. |
| No progression | N | This code was assigned when the expert considered that the policy was implemented and had no further progression, or when the starting maturity was considered to be at level 5. |
| Technology | T | This code was assigned when the expert's progression description was related to an increase in technological solutions or required more advanced technologies for the progression of the policy. |

To determine whether there was consensus on the starting maturity the mean, the mode, the sub-mode and the confidence intervals for the mean were calculated. The mode was the starting maturity with the greatest number of experts. In case it existed, the sub-mode was a maturity level with the frequency of the mode minus one (e.g., if the mode was level 1 with five experts and level 2 had four experts, level 2 would be the sub-mode). On the other hand, the confidence interval for the mean was calculated for 95% confidence. Although the distribution of the data is unknown, the confidence intervals were calculated assuming normality of the data a common assumption known to have satisfactory results even in non-normal distributions [42–44]. The mean and the confidence interval's limits where rounded in order to have integer values and, therefore, no partial maturity levels. Once these calculations were made, a decision on whether there was consensus was taken using the decision tree in Figure 1.

As shown in Figure 2, there are five possible cases for each policy, for clarity, these cases will be numbered in the following description by using N1–N5.

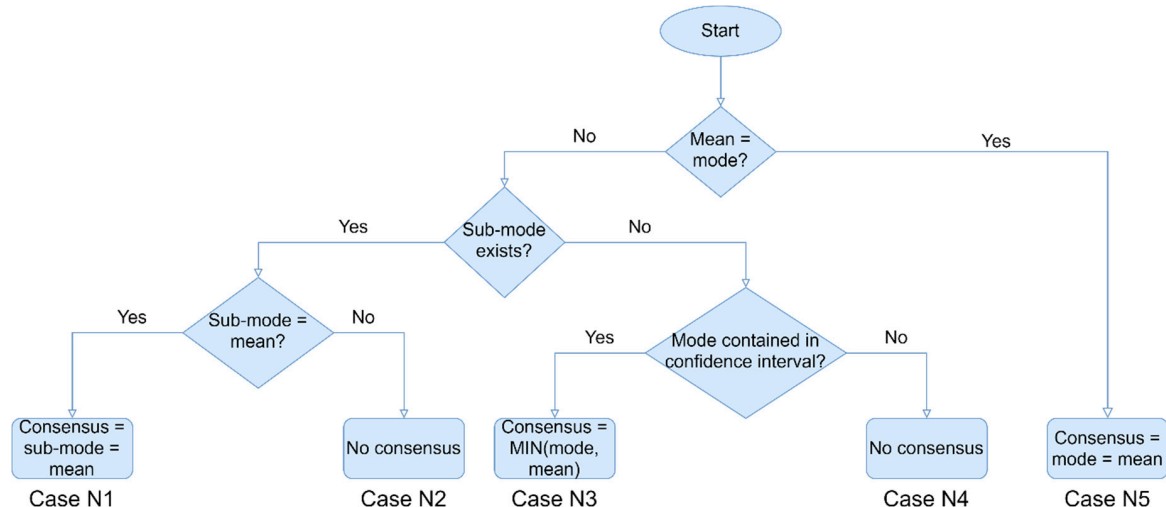

**Figure 2.** Quantitative analysis decision tree.

N1: This case was reached when the rounded mean was different to the mode; the sub-mode existed and was equal to the mean. In this case, the consensus was considered to be the sub-mode because a large group of experts considered this as the starting maturity level for the policy and, the experts who did not, were closer to this starting maturity than to the mode starting maturity.

N2: This case was reached when the rounded mean was different to the mode; the sub-mode existed but was not equal to the mean. In this case, there was no consensus because neither the mode, nor the sub-mode were around the starting maturity level where the mean expert considered the policy should start.

N3: This case was reached when the mode and the rounded mean were not equal, there was no sub-mode, and the confidence intervals (CI) contained the mode. In this case, the consensus was the minimum between the mode and the mean. This criterion was applied because the mode and the mean were theoretically not so far apart since it was in the CIs of the mean. The reason for choosing the minimum of the two is that it is more beneficial for cyber resilience building to diversify the investment in policies and to start the implementation as early as possible as suggested by previous studies [19].

N4: This case was reached when the mode and the rounded mean were different, there was no sub-mode, but the confidence intervals (CI) did not contain the mode. In this case, no consensus was reached because it meant that many experts considered one starting maturity, but that starting maturity was considerably far from most of the other experts' opinion on the policy's starting maturity.

N5: Finally, this case was reached when the rounded mean was equal to the mode. In this case, the consensus was easily reached because it meant that most experts thought that one starting maturity was predominant and that the experts who diverged from this opinion were not diverging too much from it.

In order to decide whether there was consensus on the progression types the mode and the mode's percentage of agreement were calculated. The mode's percentage of agreement was the percentage of experts who considered the mode progression type as the main progression type. This means that the mode's frequency was divided by the number of total experts, not the number of total progression types assigned to the policy because experts could describe progressions that were a combination of different progression types. If the percentage of agreement was over 50%, the progression type was considered to be the consensus. If the percentage of agreement was lower, there was no consensus for the policy's progression type.

5. Detailed case interpretations, which in this context consisted of creating a progression model based on the interpretation of the most common starting point for each policy and its most common progression type (code). In this step, when there was a tie in the starting maturity of a

policy the lower maturity was used, and when there was a tie in the progression type a mix of both progressions was used for the construction of the progression model.

A summary of the five-step methodology with the results from each step is shown in Figure 3.

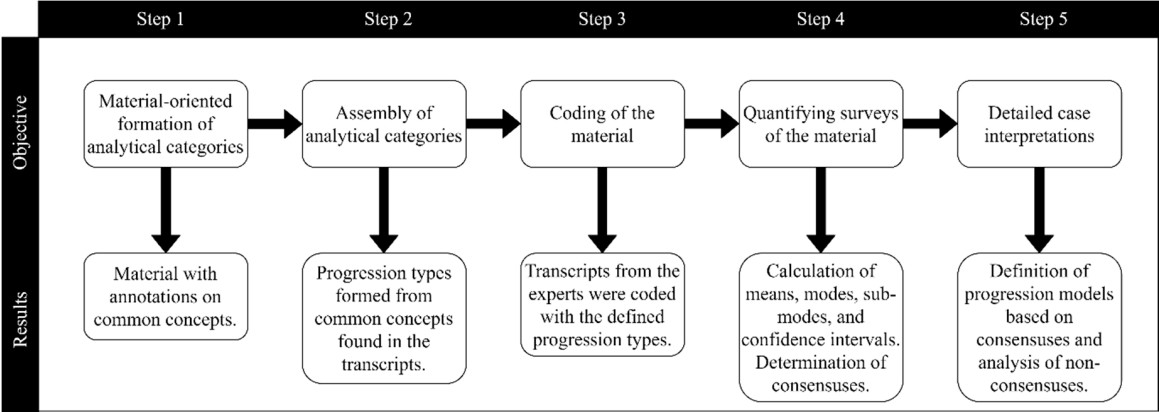

**Figure 3.** Transcript analysis methodology summary.

## 3. Results

This section will present the results obtained from applying the previously described methodology. These results are presented in different subsections corresponding to each of the cyber resilience domains. Since each domain contains several policies, three tables will be presented in each sub-section: the number of experts who considered each starting maturity level and the data used to determine the consensus (Table 2). The number of experts who described each progression type, the mode, and the percentage of agreement with that mode (Table 3). Finally, how this information can be used to construct a progression model (Table 4). The text in each subsection will briefly describe the domain and interpretation of the tables mentioned previously.

**Table 2.** Governance policies' starting maturity.

| Policy | Policy Code | 1 | 2 | 3 | 4 | 5 | Mean | Mode | Sub-Mode | LCI | UCI | Consensus |
|---|---|---|---|---|---|---|---|---|---|---|---|---|
| Develop and communicate a cyber resilience strategy. | G1 | 4 | 4 | 2 | 1 | 0 | 2 | 1;2 | N/A | 1 | 3 | 2 |
| Comply with cyber resilience-related regulation. | G2 | 5 | 2 | 3 | 1 | 0 | 2 | 1 | N/A | 1 | 3 | 1 |
| Assign resources (funds, people, tools, etc.) to develop cyber resilience activities. | G3 | 4 | 1 | 5 | 1 | 0 | 2 | 3 | 1 | 1 | 4 | No consensus |

The background highlights the maximum frequency for each policy.

**Table 3.** Governance policies' progression type.

| Policy | Policy Code | II | S | E | F | I | O | P | N | Mode | % of Agreement |
|---|---|---|---|---|---|---|---|---|---|---|---|
| Develop and communicate a cyber resilience strategy. | G1 | | 1 | 2 | 2 | | 2 | 7 | | P | 64% |
| Comply with cyber resilience-related regulation. | G2 | 1 | | 6 | 1 | | 3 | 5 | 1 | E | 55% |
| Assign resources (funds, people, tools, etc.) to develop cyber resilience activities. | G3 | 3 | | 2 | 2 | 2 | 4 | 2 | | O | 36% |

The background highlights the maximum frequency for each policy.

**Table 4.** Governance policies' progression model.

| Policy\Progression | 1 | 2 | 3 | 4 | 5 |
|---|---|---|---|---|---|
| G1 | | There is a cyber resilience strategy that centers on protecting the systems according to their risks (implement traditional cybersecurity). | The cyber resilience strategy defines resilience requirements based on the risks of the company's assets. The company tries to comply with these resilience requirements to the best of their abilities. This includes having response plans in case of incidents that could harm the compliance with these requirements. | The company's strategy is detailed and tries to go in depth on how to make the systems and processes as resilient as possible with specific plans on how to recover in case the protection methods fail. | The strategy is continuously improved upon, trying to implement lessons learned from the company's previous iterations of the strategy and previous successes or mistakes. |
| G2 | The company has identified the cyber resilience or cybersecurity related laws and regulations that directly concern their activity. | The company does its best to comply with the most directly related cyber resilience/cybersecurity laws and regulations. | The company tries to comply with the laws and regulations that have been identified by internally auditing which are being complied with and which are still in progress. | The company starts exploring laws and regulations that can indirectly concern their activity and sees added value in complying with these laws as a way to improve their cyber resilience. | The company continuously complies with more demanding regulations driven by their own cyber resilience implementation and not simply with the intention of complying. |
| G3 | | | Specific, documented budgets and resources are assigned for the fulfillment of the cyber resilience strategy. | The investments in cyber resilience are controlled through KPIs that the company has elected to try to optimize their allocation of resources. | Resources are flexibly moved in order to maximize the benefits of the resources that have been assigned and optimize the values of the company's KPIs. |

### 3.1. Governance

Governance is the cyber resilience domain that contains the policies and procedures to manage cyber resilience from a strategic point of view. Thus, this domain reflects how involved the top management of the company is with cyber resilience and how cyber resilience is managed in a company [24].

As previous research suggests, governance contains three policies, Table 2 shows these policies and the number of experts who selected each starting maturity level and the criteria used to find a consensus.

As shown in the data, the consensus was reached for the first two governance policies by using the criteria described in the methodology. For policy G1, the mean and one of the modes in the data are the same and, therefore, the consensus is reached. Likewise, in policy G2, the mean and the mode are not equal, but since there is no sub-mode, the confidence intervals (lower confidence interval limit (LCI) and upper confidence interval limit (UCI)) contain the mode and, therefore, a consensus is reached on the mode.

On the other hand, policy G3 reaches no statistical consensus since the mean is not equal to the mode and the sub-mode is not equal to the mean either. This policy does not reach a consensus because experts' are divided between what is considered resource assignment or not. The experts who argued that any resource spent in cyber resilience activities is considered to be assigned, placed the starting maturity at level 1. On the other hand, experts who considered that assigning resources meant seriously budgeting those resources and formally assigning them for cyber resilience placed the beginning of this policy at level 3.

Moreover, the analysis of the progression types for these policies is shown in Table 3. As the table shows, the modes for the progressions of G1 and G2 contain over 50% of the experts' responses. In this case, the progression types corresponding to these policies are proactivity and expansion respectively. Therefore, a consensus is reached with these progression types for these policies. However, policy G3 does not reach a consensus with only 36% of the experts agreeing with the policies' mode progression type. This non-consensus was because many experts considered this policy to progress mainly due to an investment increase, while others considered that the progression was more towards the optimization of the investments.

Considering the consensuses reached in G1 and G2 a progression model can be constructed by starting at levels 2 and 1, respectively, and using the proactivity and expansion progression types. However, in the case of policy G3, no consensus is reached on the starting point nor the progression type. By using the modes to construct a progression model for G3, the starting maturity would be level 3 and the progression type would be optimization. However, this criterion was used to provide guidelines for companies, but this does not represent a real consensus and in this policy, the criteria of the company will still be needed. The constructed progression model using these criteria and the experts' progression models that met them is shown in Table 4.

### 3.2. Risk Management

Risk management is the cyber resilience domain that defines the criteria and procedures to systematically identify, document, classify, and mitigate/accept the company's cyber risks [14,24].

In this domain, there are four main policies [24]. The general agreement between the experts is that these risk management policies should be started from level 2, as shown in Table 5. In most of these cases (RM1, RM2, and RM4) the criteria used to define a consensus is that the mean and the mode are the same. However, RM3 is a slightly different case since the mean and the mode are different. Therefore, considering that there is no sub-mode and that the mode is contained in the confidence interval, the mode is considered the consensus for being the minimum between the mode and the mean.

**Table 5.** Risk management policies' starting maturity.

| Policy | Policy Code | 1 | 2 | 3 | 4 | 5 | Mean | Mode | Sub-Mode | LCI | UCI | Consensus |
|---|---|---|---|---|---|---|---|---|---|---|---|---|
| Systematically identify and document the company's cyber risks. | RM1 | 4 | 6 | 1 | 0 | 0 | 2 | 2 | N/A | 0 | 3 | 2 |
| Classify/prioritize the company's cyber risks. | RM2 | 0 | 7 | 3 | 1 | 0 | 2 | 2 | N/A | 1 | 4 | 2 |
| Determine a risk tolerance threshold. | RM3 | 0 | 5 | 3 | 3 | 0 | 3 | 2 | N/A | 2 | 4 | 2 |
| Mitigate the risks that exceed the risk tolerance threshold. | RM4 | 1 | 5 | 4 | 1 | 0 | 2 | 2 | 3 | 1 | 3 | 2 |

The background highlights the maximum frequency for each policy.

In the case of the progression types shown in Table 6, the experts' reach a clear consensus in RM1 and RM2 as policies that evolve with an increase of formalization and more divided in RM3 and RM4.

The RM3 policy had most experts consider its progression as a formalization progression, but closely behind were the optimization, proactivity, and no evolution. All these progressions are possible because optimization of a risk tolerance threshold is related to measuring the risk quantitatively and using the threshold as a KPI that the company can optimize. Proactivity is reasonable because it is possible to start viewing the definition of a tolerance threshold as a goal and keep proactively improving that goal continuously. Thus, proactivity, optimization, and formalization are not mutually exclusive and even reasonable-to-combine types of progression. The "no progression" type, however, is a complete opposite point of view in which the experts considered that the risk tolerance threshold was a defined number that did not require any further progression. This opinion from three experts is due to a more challenging definition of a risk tolerance threshold instead of progressive smaller goals, which are more natural for the other three progression types.

On the other hand, RM4 was primarily considered to have an expansion progression type, but it was closely followed by formalization and proactivity. In this sense, most experts considered that mitigation starts from an urgency feeling and expands into the mitigation of more and more risks (expansion). Other experts, however, considered that it could also be less intuitive, more documented, more traceable mitigations (formalization) or even a less reactive and more preventive mitigation of risks (proactivity) as maturity increases.

By using the consensuses and the mode criteria in the cases where there was no consensus, Table 7 shows a progression of the risk management policies.

**Table 6.** Risk management policies' progression type.

| Policy | Policy Code | II | C | S | E | F | I | O | P | N | Mode | % of Agreement |
|---|---|---|---|---|---|---|---|---|---|---|---|---|
| Systematically identify and document the company's cyber risks. | RM1 | | 2 | | 3 | 9 | 1 | 1 | 3 | | F | 82% |
| Classify/prioritize the company's cyber risks. | RM2 | | 2 | 1 | 1 | 7 | 1 | 2 | 2 | 2 | F | 64% |
| Determine a risk tolerance threshold. | RM3 | | | | | 4 | 1 | 3 | 3 | 3 | F | 36% |
| Mitigate the risks that exceed the risk tolerance threshold. | RM4 | 2 | | | 1 | 5 | 4 | 1 | 3 | 4 | | E | 45% |

The background highlights the maximum frequency for each policy.

**Table 7.** Risk management policies' progression model.

| Policy\Progression | 1 | 2 | 3 | 4 | 5 |
|---|---|---|---|---|---|
| RM1 | | Risks are determined intuitively and according to the experience of the personnel. | A list of risks associated to the company's assets has been put together based on some research that tries to determine all the risks associated to the assets. | There is a systematic procedure used to identify all the risks associated to the company's assets. This procedure includes research, vulnerability management, etc. Risks are formally quantified according to their impact and probability. | The systematic procedure used to identify risks is repeated periodically to update the risks in the company. As much sources of information are used to identify these risks, including information from maintenance such as mean time before failure and mean time to recovery (used to calculate probability of downtime). |
| RM2 | | Identified risks are prioritized intuitively, according to the experience of the personnel and based on urgency towards the development of the company's activity. | Risks are classified and prioritized based on research of the impact they may have in the company's activity. This classification and prioritization are now documented. | Risks are calculated based on their impact and probability. The numerical risk values are considered when prioritizing risks. There is rigorous documentation of the risk associated to all the company's assets. | The systematic and formal risk classification and prioritization is updated periodically to have a realistic measure of the company's risk. |
| RM3 | | The risk tolerance threshold is put arbitrarily, mostly based on the abilities of the company's personnel to address the identified risks. | Risk tolerance is based on the possible impact of the risks. | The risk tolerance threshold is documented as a value of risk (impact x probability). | The risk tolerance threshold is continuously updated to more demanding values as the company's cyber resilience measures minimize risk. |
| RM4 | | The company mitigates some of the risks that have been identified and that affect the most important assets. | The company mitigates all the risks that affect important assets and some other risks that they can address. | The company mitigates most of the risks that exceed the risk tolerance threshold. | The company systematically mitigates all the risks (including newly discovered ones) that surpass each update of the risk tolerance threshold. |

### 3.3. Asset Management

Asset management is the cyber resilience domain that involves the management of devices, systems, software, services, and facilities that enable the organization to achieve business purposes [14].

Asset management contains five main policies [24]. These policies and their starting maturity levels are shown in Table 8. As shown in the table, there is consensus on most of these policies. In policies AM1, AM2, and AM4 the mean is equal to the mode and, therefore, there is consensus. For policy AM5, there was consensus because there was no sub-mode and the mode was contained in the confidence intervals.

**Table 8.** Asset management policies' starting maturity.

| Policy | Policy Code | 1 | 2 | 3 | 4 | 5 | Mean | Mode | Sub-Mode | LCI | UCI | Consensus |
|---|---|---|---|---|---|---|---|---|---|---|---|---|
| Make an inventory that lists and classifies the company's assets and identifies the critical assets. | AM1 | 6 | 5 | 0 | 0 | 0 | 1 | 1 | 2 | 0 | 3 | 1 |
| Create and document a baseline configuration for the company's assets. | AM2 | 1 | 4 | 5 | 1 | 0 | 3 | 3 | 1 | 1 | 4 | 3 |
| Create a policy to manage the changes in the assets' configurations. | AM3 | 1 | 4 | 2 | 3 | 1 | 3 | 2 | 4 | 2 | 4 | No consensus |
| Create a policy to periodically maintain the company's assets. | AM4 | 4 | 5 | 1 | 0 | 1 | 2 | 2 | 1 | 1 | 3 | 2 |
| Identify and document the internal and external dependencies of the company's assets. | AM5 | 1 | 5 | 3 | 2 | 1 | 3 | 2 | N/A | 2 | 4 | 2 |

The background highlights the maximum frequency for each policy.

For policy AM3 on the other hand no consensus was reached because neither the mode nor the sub-mode were equal to the mean starting maturity level. The reason for the non-consensus in AM3 is that some experts considered that companies should start implementing a change control policy in a basic form and related to the next policy (AM4) from a starting maturity of level 2. Conversely, others considered that a change control policy would require the implementation of a stable situation through a standardized base configuration of assets (AM2); therefore, these experts suggested that AM3 should be implemented at a starting maturity of 4.

Regarding the progression types of asset management policies, as shown in Table 9, there is only a consensus on AM1 having a specificity progression and AM4 having a proactivity progression. On the other hand, for configuration definitions and change control policies (AM2 and AM3) some experts consider there is a mix between formalization (documentation and systematization) and technology (increase in the complexity or number of technological solutions) but proactivity (perceiving and actively pursuing the benefits of performing these tasks) was also considered a common type of progression. In the case of AM3, some experts also argued that it could have a progression where it started for certain assets and expanded to the rest of the assets (expansion).

Finally, for AM5 experts argued that the progression of the dependency management was mostly in formalization and proactivity, meaning that dependency management was more systematic and standardized, and that companies saw more value and changed attitude towards it as their maturity increased. However, other experts had different opinions, most of the divergent opinions tended towards specificity, which referred in this case to finding assets' components and parts' dependencies. Therefore, no consensus on this policy's progression type was found.

Using the mode criteria for the starting maturity and the progression types where there is no consensus, the defined consensuses and progressions given by the experts, a progression model was constructed and is presented in Table 10.

**Table 9.** Asset management policies' progression type.

| Policy | Policy Code | C | S | E | F | I | O | P | N | T | Mode | % of Agreement |
|---|---|---|---|---|---|---|---|---|---|---|---|---|
| Make an inventory that lists and classifies the company's assets and identifies the critical assets. | AM1 | | 6 | 4 | 4 | | | 2 | | 3 | S | 55% |
| Create and document a baseline configuration for the company's assets. | AM2 | | 1 | 2 | 4 | 1 | 1 | 3 | | 4 | F;T | 36% |
| Create a policy to manage the changes in the assets' configurations. | AM3 | | 1 | 3 | 4 | 1 | 1 | 3 | | 4 | F;T | 36% |
| Create a policy to periodically maintain the company's assets. | AM4 | 2 | | 1 | 4 | 1 | 2 | 6 | 1 | 1 | P | 55% |
| Identify and document the internal and external dependencies of the company's assets. | AM5 | 1 | | 4 | 5 | 1 | 1 | 5 | | 3 | F;P | 45% |

The background highlights the maximum frequency for each policy.

**Table 10.** Asset management policies' progression model.

| Policy\Progression | 1 | 2 | 3 | 4 | 5 |
|---|---|---|---|---|---|
| AM1 | There is a list of the company's assets. | The company's inventory includes more information about the assets such as model, manufacturer, etc. | The company's inventory also includes the physical and logical location of the assets. | The inventory of the company's assets includes specific information about components in assets in which this may apply. | The company's inventory is highly specific with as much information of the assets as possible (e.g., components, make, value, location, risk value, etc.) |
| AM2 | | | An undocumented base configuration is used to set up new systems in the company. | There is a documented base configuration for the company's assets. A Configuration management database (CMDB) is used to control document the base configurations | The base configuration of the company's assets is standardized and used in all of the systems. The CMDB is automatically updated as new assets are introduced or configurations are changed. |
| AM3 | | The company controls basic changes that have been made due to corrective maintenance issues. | The company's personnel starts to control the changes needed for the informal base configuration. | The company documents and has traceability of the changes made to the base configuration of the systems through a CMDB, | The traceability of changes made to any system in the system is registered in the CMDB as soon as possible after a change to a configuration has been made and through a standard, documented procedure. |
| AM4 | | The company does corrective maintenance to its assets. | The company occasionally updates the systems. | The company periodically does preventive maintenance and tries to keep the systems up to date. | The company has a system of predictive maintenance based on previous data of mean time before failure and mean time to repair. |
| AM5 | | The main dependencies are identified because of the knowledge of the company's personnel. | There is a documented list of the identified dependencies between systems. | The dependencies are systematically identified for all of the company's assets and documented in the dependency list. | The company does its best to identify all the internal and external dependencies from all of its assets and keep the dependency list updated in order to ease the contingency/business continuity planning. |

### 3.4. Threat and Vulnerability Management

Threat and Vulnerability management is the domain that encompasses the processes used to identify, prioritize, document, and find protective methods for threats and vulnerabilities based on the current company's systems [18].

Threat and vulnerability management has 2 main policies [24], their starting maturity levels and progression types are shown in Table 11. As shown in the table, in both policies, the mean starting maturity is equal to the mode and, therefore, there is a consensus.

**Table 11.** Threat and vulnerability management policies' starting maturity.

| Policy | Policy Code | 1 | 2 | 3 | 4 | 5 | Mean | Mode | Sub-Mode | LCI | UCI | Consensus |
|---|---|---|---|---|---|---|---|---|---|---|---|---|
| Identify and document the company's threats and vulnerabilities. | TVM1 | 3 | 4 | 4 | 0 | 0 | 2 | 2;3 | 1 | 1 | 3 | 2 |
| Mitigate the company's threats and vulnerabilities. | TVM2 | 2 | 1 | 6 | 2 | 0 | 3 | 3 | N/A | 1 | 4 | 3 |

The background highlights the maximum frequency for each policy.

On the other hand, regarding progression types and as shown in Table 12 there is consensus on TVM1 having a formalization progression type, but there is no consensus for TVM2's progression. Some experts considered that TVM2 could have an optimization progression type in which the company should try to establish KPIs (such as risk quantification or percentage of patched vulnerabilities) and try to optimize these KPIs. Other experts, however, considered that TVM2 can have a progression similar to RM4 (expansion) or have a combination of formalization, optimization and proactivity which would represent a progression in which as maturity increases there is more systematization, measurement, and pursuit of continuous improvement.

**Table 12.** Threat and vulnerability management policies' progression type.

| Policy | Policy Code | II | C | S | E | F | I | O | P | N | T | Mode | % of Agreement |
|---|---|---|---|---|---|---|---|---|---|---|---|---|---|---|
| Identify and document the company's threats and vulnerabilities. | TVM1 | 1 | 2 |  | 5 | 7 | 1 |  | 6 | 1 |  | F | 64% |
| Mitigate the company's threats and vulnerabilities. | TVM2 | 2 | 2 | 1 | 3 | 3 | 1 | 4 | 3 |  | 1 | O | 36% |

The background highlights the maximum frequency for each policy.

Using the defined consensuses, the mode criteria for the non-consensus and progressions in the experts' progression models, Table 13 shows a progression model for the threat and vulnerability management policies.

**Table 13.** Threat and vulnerability management policies' progression model.

| Policy\Progression | 1 | 2 | 3 | 4 | 5 |
|---|---|---|---|---|---|
| TVM1 | | Threats and vulnerabilities are identified intuitively and according to the experience of the personnel. | There is a list of threats and vulnerabilities associated to the company's assets that has been put together based on some research in vulnerability repositories. | There is a systematic procedure (i.e., pen testing) used to identify all the threats and vulnerabilities associated to the company's assets. | The systematic procedure used to identify threats and vulnerabilities is repeated periodically to update the risks in the company. |
| TVM2 | | | Threats and vulnerabilities that are perceived as priorities are mitigated as soon as possible and other vulnerabilities are mitigated in arbitrary order. | Threats and vulnerabilities are quantified as risks and they are mitigated if they exceed the risk tolerance threshold. | All threats and vulnerabilities are mitigated (including newly discovered ones) when they exceed the latest update of the risk tolerance threshold. |

*3.5. Incident Analysis*

Incident analysis is the cyber resilience domain that involves the procedures and policies to make detailed investigations to obtain as much information as possible from suffered incidents with the objective of learning from those incidents and improving protection methods to prevent future incidents [14,24].

The incident analysis policies' starting maturities were strongly agreed upon as shown in Table 14. In this case, IA1 was considered to start at level 2; IA2 and IA4 were considered to start at level 3, and IA3 was considered to start at level 5.

**Table 14.** Incident analysis policies' starting maturity.

| Policy | Code | 1 | 2 | 3 | 4 | 5 | Mean | Mode | Sub-Mode | LCI | UCI | Consensus |
|---|---|---|---|---|---|---|---|---|---|---|---|---|
| Assess and document the damages suffered after an incident. | IA1 | 1 | 6 | 3 | 0 | 1 | 2 | 2 | N/A | 1 | 4 | 2 |
| Analyze the suffered incidents to find as much information as possible: causes, methods, objectives, point of entry, etc. | IA2 | 2 | 2 | 5 | 2 | 0 | 3 | 3 | N/A | 2 | 4 | 3 |
| Evaluate the company's response and response selection to the incident. | IA3 | 0 | 1 | 2 | 2 | 6 | 4 | 5 | N/A | 3 | 6 | 5 |
| Identify lessons learned from the previous incidents and implement measures to improve future responses, response selections, and risk management. | IA4 | 0 | 1 | 6 | 2 | 2 | 3 | 3 | N/A | 1 | 3 | 3 |

The background highlights the maximum frequency for each policy.

On the other hand, as shown in Table 15, only IA1 and IA3 had more than 50% of the experts considering that their progression type was the same as the mode. In these policies formalization and no progression were selected as their main progressions.

**Table 15.** Incident analysis policies' progression type.

| Policy | Code | II | C | S | E | F | I | O | P | N | Mode | % of Agreement |
|---|---|---|---|---|---|---|---|---|---|---|---|---|
| Assess and document the damages suffered after an incident. | IA1 | 1 | | 3 | 1 | 6 | | | 3 | 1 | F | 55% |
| Analyze the suffered incidents to find as much information as possible: causes, methods, objectives, point of entry, etc. | IA2 | | 1 | 5 | 3 | 4 | 1 | | 4 | | S | 45% |
| Evaluate the company's response and response selection to the incident. | IA3 | 1 | | 1 | | 3 | | 1 | | 6 | N | 55% |
| Identify lessons learned from the previous incidents and implement measures to improve future responses, response selections, and risk management. | IA4 | | | | 1 | 5 | | | 3 | 3 | F | 45% |

The background highlights the maximum frequency for each policy.

In the cases of IA2 and IA4, no consensus was reached for their progression type. Many experts considered IA2 to have a specificity progression, but others considered that this policy could have a formalization or proactivity progression as well. The specificity progression in this policy refers to more specific and detailed analyses of incidents when they happen (better forensics). A formalization progression in IA2 would mean more systematic and standardized approaches to the forensic analysis and finally a proactivity progression would mean that the company learns to perceive the benefits of the analysis and thus has a proactive attitude towards inquiring causes, methods, points of entry, etc., of incidents.

For policy IA4, many experts described a formalization progression, but more than half of the experts considered that the policy could progress by increasing proactivity or not progress at all once it has been implemented.

Based on the majority of the experts' opinions on the starting maturity and progression types for each of the incident analysis policies, a progression model was constructed and it is presented in Table 16.

**Table 16.** Incident analysis policies' progression model.

| Policy\Progression | 1 | 2 | 3 | 4 | 5 |
|---|---|---|---|---|---|
| IA1 | | The company informally evaluates their losses after an incident and the systems that need repairing or replacing. | The company evaluates their losses and documents them. | The company has a documented procedure to evaluate the damages caused by an incident. | The company has a documented and systematic (using the dependencies) procedure to evaluate all the systems after an incident in order to detect all of the incident's consequences. |
| IA2 | | | The company does general forensics to determine the way in which the incident happened. | The company tries to identify the methods and entry points after an incident. | The company does a full forensics evaluation in which causes, methods, and entry points are fully discovered. |
| IA3 | | | | | There is a systematic procedure to evaluate the company's response and response selection after every incident in order to improve decision-making in future incidents. |
| IA4 | | | The company learns from the information obtained from the incident analysis and thus from the occurrence of every incident. | The company uses a documented procedure to identify lessons from previous incidents based on the way their forensics analysis and damage analysis is made. | The company systematically implements the lessons learned from incidents and documents them for future reference. |

*3.6. Awareness and Training*

Awareness and training is the cyber resilience domain concerned with plans, procedures, and policies to provide cybersecurity awareness education and technical training to perform cyber resilience-related duties and responsibilities to the company's personnel [14,18,24].

For awareness and training, the 4 main policies [24] and the number of experts who placed each policy in each starting maturity are presented in Table 17. As shown by the table, a consensus on the starting points for these policies is reached by either the confidence interval containing the mode (AT1 and AT2), or by the criteria where the mode is equal to the mean (AT3 and AT4).

**Table 17.** Awareness and training policies' starting maturity.

| Policy | Policy Code | 1 | 2 | 3 | 4 | 5 | Mean | Mode | Sub-Mode | LCI | UCI | Consensus |
|---|---|---|---|---|---|---|---|---|---|---|---|---|
| Define and document training and awareness plans. | AT1 | 3 | 1 | 6 | 1 | 0 | 2 | 3 | N/A | 1 | 4 | 3 |
| Evaluate the gaps in the personnel skills needed to perform their cyber resilience roles and include these gaps in the training plans. | AT2 | 0 | 2 | 2 | 7 | 0 | 3 | 4 | N/A | 2 | 5 | 4 |
| Train the personnel with technical skills. | AT3 | 2 | 3 | 4 | 2 | 0 | 3 | 3 | 2 | 2 | 3 | 3 |
| Raise the personnel's awareness through their training programs. | AT4 | 3 | 4 | 3 | 1 | 0 | 2 | 2 | 1;3 | 1 | 3 | 2 |

The background highlights the maximum frequency for each policy.

As for progression types, as shown in Table 18, there is consensus on AT1, AT3, and AT4 with specificity, specificity, and formalization progression types, respectively. Policy AT2 did not reach a consensus since some experts considered that evaluation of gaps in skills and abilities could be repeated periodically (continuity progression) while others believed that it was done once and there was no need for periodic nor any other type of progression (no progression).

**Table 18.** Awareness and training policies' progression type.

| Policy | Policy Code | II | C | S | E | F | I | O | P | N | Mode | % of Agreement |
|---|---|---|---|---|---|---|---|---|---|---|---|---|
| Define and document training and awareness plans. | AT1 | | | 6 | 2 | 2 | | 1 | 2 | | S | 55% |
| Evaluate the gaps in the personnel skills needed to perform their cyber resilience roles and include these gaps in the training plans. | AT2 | | 4 | 2 | | 1 | 1 | | 2 | 3 | C | 36% |
| Train the personnel with technical skills. | AT3 | 1 | 1 | 6 | 1 | 2 | | 1 | 3 | | S | 55% |
| Raise the personnel's awareness through their training programs. | AT4 | | 3 | 3 | 6 | | 1 | 1 | 2 | | F | 55% |

The background highlights the maximum frequency for each policy.

Using these consensuses and the mode criteria for the non-consensuses, Table 19 presents a progression model for the awareness and training policies.

**Table 19.** Awareness and training policies' progression model.

| Policy\Progression | 1 | 2 | 3 | 4 | 5 |
|---|---|---|---|---|---|
| AT1 | | | There is a general cyber resilience training plan for all the employees in the company. | There are plans defined according to different profiles of the employees. | Each employee has training plans according to their needs and gaps in abilities. |
| AT2 | | | | The company evaluates the gaps in the personnel abilities to perform their cyber resilience tasks in order to define the training plans. | The company periodically evaluates the gaps in the personnel's knowledge and abilities in order to keep the plans updated. |
| AT3 | | | The technical personnel receives general technical training. | All the personnel receives technical training needed according to their profile and general tasks performed by employees from that profile. | All the personnel receives technical training according to their specific (personal) needs and gaps in their abilities. |
| AT4 | | There are undocumented and/or unfollowed cyber resilience rules for everyone related to their awareness. | There are occasional awareness communications for basic cyber resilience measures in which all the employees can participate. | There are periodical (with a defined period), documented and planned awareness training sessions or communications in the company. | The company systematically and periodically does awareness training courses or communications for the employees such as spam exercises, training sessions, etc. |

## 3.7. Information Security

Information security is the cyber resilience domain that involves measures and procedures to maintain confidentiality, integrity, and availability of the company's assets and information [24].

Information security policies and the number of experts who placed them in each starting maturity level are shown in Table 20. As shown in the table, the starting maturities for IS1, IS2, and IS3 are defined by a consensus on levels 1, 2, and 1, respectively. IS1 and IS3 have a defined consensus because of the confidence interval criterion, and IS2 has a consensus because the mode is equal to the mean starting maturity.

**Table 20.** Information security policies' starting maturity.

| Policy | Policy Code | 1 | 2 | 3 | 4 | 5 | Mean | Mode | Sub-Mode | LCI | UCI | Consensus |
|---|---|---|---|---|---|---|---|---|---|---|---|---|
| Implement measures to protect confidentiality (e.g., access control measures, network segmentation, cryptographic techniques for data and communications, etc.) | IS1 | 6 | 2 | 3 | 0 | 0 | 2 | 1 | N/A | 0 | 3 | 1 |
| Implement integrity checking mechanisms for data, software, hardware and firmware. | IS2 | 4 | 4 | 3 | 0 | 0 | 2 | 1;2 | 3 | 1 | 3 | 2 |
| Ensure availability through backups, redundancy, and maintaining adequate capacity. | IS3 | 5 | 3 | 3 | 0 | 0 | 2 | 1 | N/A | 1 | 3 | 1 |

The background highlights the maximum frequency for each policy.

In the case of the information security domain there was also clear consensus on the progression type for these policies. As shown in Table 21, most experts considered these policies to have mostly a technological progression.

**Table 21.** Information security policies' progression type.

| Policy | Policy Code | II | S | E | F | O | P | T | Mode | % of Agreement |
|---|---|---|---|---|---|---|---|---|---|---|
| Implement measures to protect confidentiality (e.g., access control measures, network segmentation, cryptographic techniques for data and communications, etc.) | IS1 | | 5 | 3 | 4 | | 1 | 8 | T | 72% |
| Implement integrity checking mechanisms for data, software, hardware and firmware. | IS2 | | 3 | 3 | 4 | 1 | 2 | 7 | T | 64% |
| Ensure availability through backups, redundancy, and maintaining adequate capacity. | IS3 | 1 | 3 | 5 | 4 | | 1 | 7 | T | 64% |

The background highlights the maximum frequency for each policy.

Considering the experts' consensus on the starting maturity levels and progression types for information security policies, Table 22 presents a progression model for this domain.

**Table 22.** Information security policies' progression model.

| Policy\Progression | 1 | 2 | 3 | 4 | 5 |
|---|---|---|---|---|---|
| IS1 | The company has basic measures to protect confidentiality (e.g., access control for computers and systems) | The company has implemented permission levels into the network and systems, and the access control is both physical and digital. | The company has a rigorous control of who can access the data and registers when someone has accessed it, from where, what that user has done, etc. | The company uses cryptographic techniques to give another protection level to the company's most important data. | Both stored data and communications are automatically encrypted to ensure confidentiality. |
| IS2 | Integrity measures are the same as confidentiality measures for the company at the moment. | | | Redundant data automatically double checks the integrity of the information after each modification of a register to ensure that it has not been tampered with. | The company has implemented integrity checking mechanisms such as checksums, digital certificates, block chain databases, etc. To ensure that the most important data and communications is not tampered with. |
| IS3 | The company has basic measures to ensure availability, but mainly a backup to restore availability in case of an incident. | Manual backups are made periodically of the information in all of the systems of the company and stored in hard drives disconnected from the network. | The most important data in the company is automatically backed up into several redundant copies outside the network. | There are redundancies for the most important systems in order to ensure the availability of these systems. | The company has the most advanced availability measures such as redundant high availability data processing centers with hot swapping techniques or other multiple copy methods that ensure that the data is always available. |

## *3.8. Detection Processes and Continuous Monitoring*

Detection processes and continuous monitoring involves all the processes to follow in order to actively detect cyber incidents and monitor assets to identify cybersecurity events and verify the effectiveness of protective methods [14].

For the starting maturity levels for both detection processes and continuous monitoring's policies [24] there was strong consensus from the experts as shown in Table 23. In both cases the consensus was obtained because the mean starting maturity was equal to the mode. As shown in the table, the consensus was that DPM1 started at level 2 and that DPM2 started at level 3.

**Table 23.** Detection processes and continuous monitoring policies' starting maturity.

| Policy | Policy Code | 1 | 2 | 3 | 4 | 5 | Mean | Mode | Sub-mode | LCI | UCI | Consensus |
|---|---|---|---|---|---|---|---|---|---|---|---|---|
| Actively monitor the company's assets (e.g., by implementing controls/sensors, IDS, etc.) | DPM1 | 1 | 7 | 3 | 0 | 0 | 2 | 2 | N/A | 0 | 4 | 2 |
| Define a detection process that specifies when to escalate anomalies into incidents and notifies the appropriate parties according to the type of detected incident. | DPM2 | 0 | 2 | 6 | 3 | 0 | 3 | 3 | N/A | 2 | 5 | 3 |

The background highlights the maximum frequency for each policy.

On the other hand, regarding progression types, there was no consensus on either of the policies as shown in Table 24. As shown in the table the experts were quite divided on the types of progressions for these policies. In the DPM1 policy, most experts considered that there was a mix of a technological and expansion progressions, this means that it starts with the monitorization of certain assets and expands to the rest progressively while also improving the technological solutions used to monitor the assets as maturity increases (which reflects its technological nature). However, other experts considered that the progression could be through the optimization of the monitorization, the specificity of what is monitored or the proactivity in which these assets are monitored.

**Table 24.** Detection processes and continuous monitoring policies' progression types.

| Policy | Policy Code | C | S | E | F | O | P | T | Mode | % of Agreement |
|---|---|---|---|---|---|---|---|---|---|---|
| Actively monitor the company's assets (e.g., by implementing controls/sensors, IDS, etc.) | DPM1 | 1 | 2 | 5 | 1 | 3 | 2 | 5 | E;F | 45% |
| Define a detection process that specifies when to escalate anomalies into incidents and notifies the appropriate parties according to the type of detected incident. | DPM2 | | | 2 | 5 | 1 | 3 | 3 | F | 45% |

The background highlights the maximum frequency for each policy.

DPM2 was considered by most experts to have a formalization progression, but several experts also argued about a technological progression (using fully-automatic detection notifications and processes) or a proactivity progression (finding more value in the definition of an effective detection process).

Therefore, considering the consensuses and the mode criteria for non-consensuses, Table 25 shows a progression model for the policies in the detection processes and continuous monitoring domain.

**Table 25.** Detection processes and continuous monitoring policies' progression model.

| Policy\Progression | 1 | 2 | 3 | 4 | 5 |
|---|---|---|---|---|---|
| DPM1 | | The company monitors some indicators (e.g., availability, workload, etc.) from the most important assets. | The company starts to monitor more indicators for the most important assets and starts to expand the number of assets monitored. | The company monitors most of its assets by monitoring several indicators from them. There is an alarm system that automatically detects anomalous behaviors. | The company has a complete picture of the company's operations from the monitorization of several indicators in all of the company's assets and an automatic alarm system when there is anomalous behavior. |
| DPM2 | | | There is a basic, undocumented plan to call the corresponding parties when there is an incident (e.g., call IT). | There is a documented plan with clear instructions on what to do when there is an incident in the company. | There is a documented plan with clear instructions on what to do, how to communicate and to whom when there is an incident in the company. |

## 3.9. Business Continuity Management

Business continuity management is the cyber resilience domain that involves policies to define, document, test, and implement plans to maintain and restore initial functionality during and after a cyber incident [18,24].

In the domain of business continuity management, experts strongly agreed that BCM1 and BCM2 should start at maturity level 3 and that BCM3 should start at maturity level 4, as shown in Table 26. On the other hand, testing these response and recovery plans (BCM3) does require the plans to be defined and strategies on how to test them. Additionally, testing response and recovery plans often requires stopping activities to make a cyber exercise in which the plans are tested, and all of this requires more maturity. Hence, BCM3 was considered to have a starting maturity level of 4 by consensus.

**Table 26.** Business continuity management policies' starting maturity.

| Policy | Policy Code | 1 | 2 | 3 | 4 | 5 | Mean | Mode | Sub-Mode | LCI | UCI | Consensus |
|---|---|---|---|---|---|---|---|---|---|---|---|---|
| Define and document plans to maintain the operations despite different scenarios of adverse situations. | BCM1 | 0 | 3 | 6 | 2 | 0 | 3 | 3 | N/A | 1 | 4 | 3 |
| Define and document plans to respond to and recover from incidents that include recovery time objectives and recovery point objectives. | BCM2 | 1 | 1 | 8 | 1 | 0 | 3 | 3 | N/A | 1 | 5 | 3 |
| Periodically test the business continuity plans to evaluate their adequacy and adjust them to achieve the best possible operations under adverse situations. | BCM3 | 0 | 0 | 1 | 9 | 1 | 4 | 4 | N/A | 2 | 6 | 4 |

The background highlights the maximum frequency for each policy.

For this domain, policies BCM1 and BCM2 also had a consensus on the progression type as shown in Table 27. These two policies progressed by a combination of expansion and formalization. The combination present in both response and recovery plans (BCM1 and BCM2) means that the planning for response and recovery can start for some assets (usually the critical assets) and expand

to other assets as the maturity increases (expansion), but these plans can also become more formal, standardized, documented, and systematic (formalization).

**Table 27.** Business continuity management policies' progression type.

| Policy | Policy Code | II | C | S | E | F | O | P | N | Mode | % of Agreement |
|---|---|---|---|---|---|---|---|---|---|---|---|
| Define and document plans to maintain the operations despite different scenarios of adverse situations. | BCM1 | | | | 6 | 6 | 1 | 3 | | E;F | 55% |
| Define and document plans to respond to and recover from incidents that include recovery time objectives and recovery point objectives. | BCM2 | | | | 6 | 6 | 1 | 2 | 1 | E;F | 55% |
| Periodically test the business continuity plans to evaluate their adequacy and adjust them to achieve the best possible operations under adverse situations. | BCM3 | 1 | 3 | 1 | 1 | 1 | 1 | 2 | 3 | C;N | 27% |

The background highlights the maximum frequency for each policy.

On the other hand, there was no consensus on how the BCM3 policy progressed since experts considered a wide variety of possible progressions for this policy including no progression and continuity progressions. These completely opposite progression types are due to the advanced level of maturity required for this policy since this made some experts consider that there was no need for more progression while others considered that although it was advanced, testing continuity plans could be done periodically.

Considering the above-mentioned starting maturity levels and progression types (either by consensus or the mode criteria), Table 28 shows a progression model for the business continuity management policies.

### 3.10. Information Sharing and Communication

Information sharing and communication is the cyber resilience domain that involves the policies and procedures to communicate the appropriate information to the appropriate parties in each moment [24]. This includes collaboration and communication with external entities and stakeholders, and the internal communication of the company in normal situations and in emergencies [24].

As shown in Table 29, the consensus among the experts was that information sharing and communication policies should start at level 3. The main argument for this starting level is that they require more maturity than many other cyber resilience policies because they are more strategic, and require systematic processes or at least more knowledge about the implementation of other policies to be able to share or benefit from sharing with other entities.

**Table 28.** Business continuity management policies' progression model.

| Policy\Progression | 1 | 2 | 3 | 4 | 5 |
|---|---|---|---|---|---|
| BCM1 | | | The company's personnel has in mind what to do in order to maintain operations of certain assets in case of certain incidents. | The company documents plans in order to protect the main assets in case of incidents. | There is a documented plan for the company's assets maintenance of operations in case of any type of incident. Plans to withstand maintenance failures are also taken into account and these failures are measured with the mean time before failure. |
| BCM2 | | | The company's personnel has in mind what to do in order recover from certain types of incidents. | The company documents plans in order recover operations in case of the major types of incidents. | There is a documented plan for the company's recovery of operations in case of any type of incident. Plans to recover from maintenance failures are also taken into account and consider the mean time to repair. |
| BCM3 | | | | Business continuity plans are tested in order to determine their effectivity in the situations they are meant to be used. | Business continuity plans are tested periodically in order to improve them and check that they are still useful despite small changes that may have happened in the company during the assigned period. |

**Table 29.** Information sharing and communication starting maturity.

| Policy | Policy Code | 1 | 2 | 3 | 4 | 5 | Mean | Mode | Sub-mode | LCI | UCI | Consensus |
|---|---|---|---|---|---|---|---|---|---|---|---|---|
| Define information sharing and cooperation agreements with external private and public entities to improve the company's cyber resilience capabilities. | SHC1 | 0 | 0 | 5 | 3 | 3 | 4 | 3 | N/A | 3 | 5 | 3 |
| Define and document a communication plan for emergencies that takes into account the management of public relations, the reparation of the company's reputation after an event, and the communication of the suffered incident to the authorities and other important third parties. | SHC2 | 1 | 1 | 6 | 3 | 0 | 3 | 3 | N/A | 2 | 4 | 3 |
| Establish collaborative relationships with the company's external stakeholders (e.g., suppliers) to implement policies that help each other's cyber resilience goals. | SHC3 | 1 | 1 | 6 | 3 | 0 | 3 | 3 | N/A | 2 | 4 | 3 |

The background highlights the maximum frequency for each policy.

On the other hand, there was no consensus for the progression types of policies SHC1 and SHC2 as shown in Table 30. In the case of SHC1 the non-consensus was due to many experts considering that cooperation agreements can have no evolution after they are defined (no progression), while many others considered that the main progression was in formalization (e.g., by documenting these agreements) and proactivity (seeing more value in the definition of the cooperation agreements as maturity increases). In the case of SHC2, there was no consensus because many experts considered that the specificity of the communication plans was the main progression (the level of detail on how the communications should be done, to whom and for what kind of incident) while others considered that formalization (systematization and documentation of the plans) could also be argued for.

**Table 30.** Information sharing and communication progression type.

| Policy | Policy Code | S | E | F | I | P | N | T | Mode | % of Agreement |
|---|---|---|---|---|---|---|---|---|---|---|
| Define information sharing and cooperation agreements with external private and public entities to improve the company's cyber resilience capabilities. | SHC1 | | 1 | 4 | | 4 | 3 | | F;P | 36% |
| Define and document a communication plan for emergencies that takes into account the management of public relations, the reparation of the company's reputation after an event, and the communication of the suffered incident to the authorities and other important third parties. | SHC2 | 5 | 2 | 4 | | 2 | | 1 | S | 45% |
| Establish collaborative relationships with the company's external stakeholders (e.g., suppliers) to implement policies that help each other's cyber resilience goals. | SHC3 | 3 | | 6 | 1 | 3 | 1 | | F | 55% |

The background highlights the maximum frequency for each policy.

In the case of SHC3, there was consensus that the main progression type for the cooperation with external stakeholders is by formalizing these agreements.

Table 31 shows a progression model for the information sharing and communication policies considering the starting maturities and the progression types defined either by consensus or the mode criterion.

**Table 31.** Information sharing and communication policies' progression model.

| Policy\Progression | 1 | 2 | 3 | 4 | 5 |
|---|---|---|---|---|---|
| SHC1 | | | Some informal relationships with other entities are established mainly because of personal contacts from the personnel. | There are documented, formal and well-defined relationships between the company and some external entities to share information about cyber resilience. | The company actively seeks to establish more formal information sharing and cooperation relationships with external entities. |
| SHC2 | | | There is a general resilience communication plan. In case any incident happens, this plan is activated. | The emergency communication plan differs in some cases depending on the type of incident that is suffered. | There are emergency communication plans defined that correspond to the possible incidents that the company may suffer (i.e., to the risks and response plans). |
| SHC3 | | | Some informal relationships with the company's providers are established mainly because of personal contacts from the personnel. | There are documented, formal and well-defined relationships between the company and some external stakeholders to cooperate and follow certain guidelines about cyber resilience. | The company actively seeks to establish more formal cooperation relationships with external stakeholders. These relationships seek to secure the supply chain as much as possible. |

## 4. Discussion

The results presented in this paper are a compilation of opinions from experts of three different profiles related to the operationalization of cyber resilience. Their background and experience have most likely forged their ideas on the ways in which cyber resilience can be built from the ground up in a company. This experience and their years as practitioners implementing cyber resilience in their current or past companies make the progression models and progression types viable for a realistic application of the policies instead of a theoretical approach. In this sense, these results can be directly applicable by companies and, therefore, be used as a guide for companies starting to operationalize cyber resilience.

Although the current literature presents frameworks and even maturity models that are meant to aid companies in the implementation of cyber resilience policies, most of the current literature fails to present the dependencies and complex relationships between these policies [14,21], much less a progression from simple to complex within the policies themselves. In other words, while the current literature is very specific, extensive and exhaustive on "which" policies to implement there is rather scarce information about how to prioritize this information (domains, policies, actions, processes, etc.) to operationalize cyber resilience.

Previous research has also tried to solve this by using an implementation order approach [39], through which the interdependencies and relationships between cyber resilience policies can be better understood and can serve as a guide for companies to start the cyber resilience building process. However, by giving natural progressions for each policy, these relationships can be understood in a richer and more complex context because progressions do not have to be set to a single state for each policy but rather have simpler and more complex versions that can relate to the simpler and more complex versions of other policies as well. The identification of an implementation order for the cyber resilience policies is helpful to aid companies in their prioritization of the investments and as shown with the results of this study it is not contradictory but complementary to defining the natural progressions for each policy. This means that using both a suggested implementation order as a guide and a progression model, companies can better understand how to use cyber resilience policies to systematically operationalize cyber resilience guided by the experience of practitioners. In other words, using the progression model can help company managers understand how each of the cyber resilience policies can manifest at different maturity states or implementation states. This should help them

strategize more effectively since they can start a systematic process to implement their desired state and later progress in the most common or natural way that the specific policy they are implementing evolves over time. Thus, the usage of a progression model in the implementation of cyber resilience policies can help companies, especially SMEs, by diminishing the need for previous knowledge and the needed experience in order to operationalize cyber resilience.

Therefore, the results of this paper can be especially significant for SMEs or other companies with low experience levels with cyber resilience or cybersecurity operationalization. For these companies, as discussed before, the best strategy would be to combine these results with previous research on cyber resilience operationalization (such as an implementation order) and then systematically evaluate their current maturity level for each policy and start aiming for a higher maturity by progressing as the experts have considered that the policy evolves. For instance, if a company is starting their cyber resilience operationalization and wants to start by making an inventory (AM1), they should first consider what they have already done, if they do not have any kind of inventory, creating a list of the assets would be a good start (as suggested in the results in Table 10). However, if they already have a list, they should either follow the rest of the example in Table 10 or, better yet, challenge themselves and keep asking themselves how they can make what they have more and more detailed since the progression for this policy was considered to be mainly in specificity.

The problem addressed in this article is the current difficulty of operationalization of cyber resilience in companies, who need to be more cyber resilient in order to thrive in the current scenario. In this line, the main goal was to develop a tool in the form of a progression model that could ease up the process of operationalizing cyber resilience in these companies. As shown in the results and the previously discussed observations, the progression model as a tool for aiding companies in cyber resilience operationalization is complementary to the current literature. Moreover, it could potentially aid companies in the strategic planning of cyber resilience implementation by helping prioritize and give insights on how the policies that are being implemented should start and progress over time. This was also achieved by interviewing practitioners with vast experience in cyber resilience operationalization (both organization practitioners and cybersecurity providers) and academic points of view. Therefore, the progression model presented in this article should be grounded in reality with the insight of experience, but also novel and in line with the current research on the field which should make the progression model both understandable for practitioners and useful to find areas of improvement or starting points to operationalize cyber resilience policies.

Having said this, the progression presented in this article are by no means the only possible ways of progressing as maturity advances. These are examples taken from the progressions the experts gave in the interviews that take into account the consensus on the starting maturity and the most common progression type. However, companies are discouraged from naïvely following any framework, guideline, maturity model or any other kind of document. Cyber resilience is a prudential competency rather than a technical ability, which means that there is no silver bullet or one-size-fits-all solution [14]. Context and circumstances always need to be taken into account and be considered in the decision-making processes that lead to the implementation of the cyber resilience policies.

On the other hand, the fact that circumstances need to be taken into account and that the progression model in this article is an example in a set of many possible results, especially in the policies where no consensus was reached. However, this does not detract from the contribution of being a guide or a starting point for companies with less experience and knowledge. On the contrary, these results are a way of attempting to eliminate variability to create the starting point; but as maturity and knowledge develop in the companies, these results should be surpassed and the experience, knowledge, and already-built cyber resilience should be the drive for these companies to keep improving their cyber resilience capabilities.

Finally, although these results contribute and complement the current literature and can aid companies in the implementation of cyber resilience, they still have limitations. On the one hand, the 11 experts have wide experience in cyber resilience implementation but future lines of research

should seek to increase the sample size and explore more opinions in order to cover more ground in the spectrum of possible progressions to adapt to even more companies' realities. This includes trying to find consensus where, in this case, has not been found and trying to find whether cultural background could influence these results. On the other hand, although these results are based on realistic applications rather than a theoretical approach, these results are still mainly theoretical and could be applied through action research or case studies to iteratively improve upon the way of transmitting this knowledge to a real-world scenario.

## 5. Conclusions

In order to survive the current cyber scenario, companies require cyber resilience. But cyber resilience is not easy to operationalize. This difficulty has led to several approaches to easing the cyber resilience operationalization but most of them list cyber resilience policies with no means of prioritizing them. This article's goal was to propose a cyber resilience progression model as a tool for companies to have realistic examples of how the essential cyber resilience policies manifest at their beginning stages and how they progress over time in the experience of practitioners and researchers. The results of this article are the starting maturity levels on a scale of 1–5 and the progressions for the 33 cyber resilience policies established in the literature as the essential policies for starting the cyber resilience operationalization process.

Therefore, the results presented in this article can be significant as they complement the current literature in aiding companies starting their cyber resilience operationalization in a more effective way. This can be achieved by letting companies strategize based on concrete descriptions that might reflect their current reality or at least let them attempt to achieve those general realities represented in each stage of the progression model.

Moreover, in order to address the limitations of this study, future lines of research should explore the inclusion of more experts to reaffirm the validity of the results and the application of these results in real scenarios in order to polish them through the richness of the empirical application.

**Author Contributions:** Conceptualization: J.F.C., S.A., and J.H.; methodology: J.F.C., S.A., L.L. and J.H.; investigation: J.F.C., S.A., and J.H., writing—original draft preparation: J.F.C., S.A., and J.H.; writing—review and editing: J.F.C., S.A., L.L. and J.H.; visualization: J.F.C., S.A., L.L. and J.H.; supervision: S.A. and J.H.; funding acquisition: S.A. and J.H. All authors have read and agreed to the published version of the manuscript.

**Funding:** This research was funded by the Basque Government grants: ELKARTEK 2020 KK-2020/00054.

**Conflicts of Interest:** The authors declare no conflict of interest. The funders had no role in the design of the study; in the collection, analyses, or interpretation of data; in the writing of the manuscript; or in the decision to publish the results.

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
