# Peer review of "Cyber Resilience Progression Model"

_applsci, doi:10.3390/app10217393_

Round 1

Reviewer 1 Report

This is a well-written paper that addresses a question of great practical relevance: how to build more cyber-resilient enterprises in practice. While there are many frameworks and maturity models, they can be quite difficult to implement, and the paper convincingly argues why practitioners might require more guidance in this respect. The methodology is clearly described and seems appropriate. The discussion of results is mature and modestly underscores limitations when appropriate.

I only have a few minor issues that ought to be addressed:

  • A confidence interval requires assumptions about the underlying distribution. This must be clarified.
  • In table 10, MTBF is mentioned as a tool appropriate for AM4. It is somewhat surprising that (i) MTBF does not also feature in the BCM section, (ii) that MTTR is not also mentioned and (iii) that the combination of MTBF and MTTR into a probability of downtime is not mentioned e.g. in the RM part. If there is any more material related to these issues in the transcribed interviews, that would be interesting to include in the paper as well. If there is not, perhaps it is worth discussing this absence in the discussion.
  • The paper covers 10 resilience domains (Section 3.1 -- 3.10). Having read them all, I think it is fair to say that this is a lot of material. Did this influence the data collection? Were there an signs of informant fatigue throughout the interviews? Were there any measures taken to prevent it? Were questions asked in the order of Section 3, or in random order to mitigate the risks of all informants being fatigued by the time they reached SHC?
  • What does the [detect] within the quote on p. 1 mean? Words within brackets are normally used to clarify terms used or to rectify grammar, but I don't see how these cases could apply here. If the source omitted detection, it seems to me that it should simply remain omitted in the quote.
  • The figures must be in higher resolution in the final version.

Typos:

p. 2 in a certain entity of field -> in a certain entity or field

p. 2-3: The paragraph on the structure of the rest of the article consists of fragmentary sentences without verbs. I suggest revising removing parentheses and adding verbs, e.g.: progression model (Section 3) -> progression model is given in Section 3

p. 3 a mean of data collection -> a means of data collection

p. 3 lack on the most recent advances -> lack the most recent advances

p. 3 vary from 3-6 maturity levels -> vary from 3 to 6 maturity levels

p. 3 in level three -> at level three

p. 3 the policy manifest -> the policy manifests

p. 3 of level three -> at level three

p. 4 There is a track-changes marking left in the heading of Section 2.2.

p.4 Interviews’ transcripts analysis -> Analysis of interview transcripts

p. 5 for a 95% of confidence -> for 95% confidence

p. 5 Once these calculations were 184 made, [MISSING END OF SENTENCE]

p. 5 As shown in [REMOVE LINEBREAK] Figure 1

p. 6 the mode was and the mode’s percentage of agreement were calculated -> the mode and the mode’s percentage of agreement were calculated

p. 6 will present -> will be presented

p. 6 high management -> top management

p. 7 the mean and the mode are not equals -> the mean and the mode are not equal

p. 8 the company has selected to try to optimize -> the company has elected to try to optimize

p. 13 Error! Reference source not found.

p. 13 measurement and a pursue of continuous improvement -> measurement and pursuit of continuous improvement

p. 14 in Error! Reference source not found.

p. 18 in Error! Reference source not found.

p. 19 Both, stored data -> Both stored data

p. 25 both, a suggested -> both a suggested

Ref 28: Conference name missing.

Author Response

Thank you very much for your time and feedback on our article. We sincerely think your input is valuable and helped us improve our research. We have copied your comments and responded to each one in the attached document.

Reviewer 2 Report

The authors deal with an interesting topic of developing a progression model to help companies strategize and prioritize cyber resilience policies by proposing the natural evolution of the policies over time. The approach is interesting as the basic idea is to make the current IT environment in firms more resilient. Although the topic is interesting, several minor insufficiencies need to be improved. These insufficiencies can be summed up in a lack of clarity.

Suggestions for improvement:

  • The manuscript should be set according to the Journal’s template and instruction to authors (text, figures, tables, equations, references, etc.). For example, the figures are unreadable, missing references, etc.
  • Check and improve the English language and grammar throughout the paper (check misspellings, writing in the first person, etc.), as well as all figures and tables (both must be readable)
  • The introduction does not provide sufficient background and includes all relevant references. The used references are not novel nor based on previous scientific papers. Also, some fundamental references are missing as well as the recent ones considering the research problem. The authors should be consistent in writing. The research problem is not clear while research goals and hypotheses are not clearly stated
  • The literature review should be improved. At the moment this section lacks a critical overview of the other approaches in solving the stated research problem and the methodology upgrade that is proposed by this research
  • The research design is not clearly written. The research methodology should be clear and the hows and the whys of used methods should be clearly visible. The methodology section is hard to follow. Some graphical overview of the proposed progression model would help readers tremendously
  • Explanations of results and their discussion are solid, but the discussion about the research significance is missing. It is not clear how the proposed procedure is validated. Also, add some additional discussion of findings in relation to the research framework as well as research goals and hypotheses are needed
  • The authors are urged to draw conclusions that are more specific. At the moment it seems like good observations and arguments that are currently missing from the discussion section. There should be a clear connection with the research problem, goals, and results

Overall, I strongly urge the authors to reconsider the above-mentioned comments, rewrite the paper accordingly, and resubmit.

Author Response

Thank you very much for your time and feedback. You have been very insightful on how to improve each section of our article with your specific suggestions. We have done our best to improve our article based on these suggestions. We attach the reproduction of your comments with our response and the changes we have made based on them.

Round 2

Reviewer 2 Report

The authors made revised version according the previously stated comments. Paper is very sound and appropriate to be puboished.